# Effectiveness of Digital Storytelling in Teaching Economics

Jana Nunvarova [1], Petra Poulova [2,*], Pavel Prazak [2] and Blanka Klimova [2]

1   Faculty of Education, University of Hradec Kralove, 500 03 Hradec Kralove, Czech Republic;
    jana.nunvarova@uhk.cz
2   Faculty of Informatics and Management, University of Hradec Kralove, 500 03 Hradec Kralove, Czech Republic;
    pavel.prazak@uhk.cz (P.P.); blanka.klimova@uhk.cz (B.K.)
*   Correspondence: petra.poulova@uhk.cz

**Abstract:** Digital storytelling is one of the teaching methods that aims to improve motivation of students, critical thinking and learning outcomes. The results of previous research show the successful use of this method in education, but its use in some subjects is still questionable. The aim of the pedagogical experiment, which was conducted in 2021 at six business academies in the Czech Republic, was to discover whether or not digital storytelling contributes to better study results in business subjects taught at high school. A total number of 856 students were randomly divided into two independent groups. In the experimental group, the digital storytelling method was used in the process of teaching. The students in the control group were taught with the standard teaching method—the teacher's explanation with the support of the presentation. By comparing the results from the pre-tests and post-tests of the experimental and control groups, the findings reveal that the students from the experimental group reached higher mean values in the post-test than the students from the control group did.

**Keywords:** digital storytelling; economics subject; education methods; students' results

## 1. Introduction

Information technology has been increasingly affecting human society. Manuel Castells argues that we live in a new historical period called "the information age", in which the life of society is gradually being transformed by new technological means [1]. An ever-denser information network connects the whole world, offers new ways of communication, and thus fundamentally changes information flows [2]. These changes affect people's behavior and their perceptions of the environment. Pictorial messages are often preferred to spoken word or script. Changes in social communication have been reflected in all areas of life, including pedagogical communication. Nowadays, information technology is considered one of the important educational tools commonly used at all levels of school education [3–5]. Short video clips related to the topic of the lesson can be used as motivational tools, electronic presentations and electronic textbooks serve as a support for teachers to introduce new topic, and Kahoot, Socrative and other applications can help to consolidate and verify students' knowledge. With the development of digital technologies, other tools and methods can possibly be used in the educational process. This includes, for example, "Educational digital storytelling", abbreviated as 'EDS', which seeks to increase students' interest in a subject and to introduce new technical terms through digital stories.

Stories can take the form of a presentation of pictures or photographs, a video, an animated film accompanied by text, a narrator's voice, or music. Engaging the listener in the story, evoking emotions, and connecting audio and visual perception can help to improve an understanding of the subject matter. Most of the investigations were conducted in humanities and social sciences lessons [6], but a pedagogical experiment focused on the impact of digital stories in STEM education came to similar conclusions [7], i.e., in science, technology, technology and mathematics [8,9]. In their study in 2019, Lestari et al. described

a pedagogical experiment that researched student motivation and the atmosphere during the EDS instruction in economics courses [10]. However, this experiment did not focus on student achievement.

Based on previous research, the authors of this study designed a pedagogical experiment to investigate whether or not digital storytelling contributes to better learning outcomes in an economic course at a secondary school in the Czech Republic. The research hypothesis was defined as follows:

**Hypothesis (H1).** *The use of educational digital storytelling leads to an increase in the individual students' results.*

The results of the pedagogical experiment show that the students from the experimental group reached higher results than the students from the control group did, and hypothesis H1 was confirmed. At the end, a questionnaire survey was conducted. Its results showed the higher motivation and interest of students in the subject and confirmed the results of previous research focused on economic subjects [10].

This paper is divided into several sections. The first two chapters explain the basic concepts of digital storytelling and introduce the elements that should not be missing in any story. The next two chapters focus on digital storytelling in education and the results from previous research studies. After an overview of the current state of the field, the methodology of the actual research and the results of the pedagogical experiment are described. The article concludes with a comparison of the results obtained with the findings of previous research and a final summary of the topic.

## 2. Literature Review

### 2.1. Storytelling

The English name "storytelling" can be translated as a story or a narrative. The method of presenting the story is not specified. For example, it can be told by a professional storyteller, by a teacher, through a theater performance, drawing or dance. Authors, in their books [11–14], explain not only the basic concepts, but also categorize the stories according to the way they are presented and their purpose, and recommend the structure of the narrative and the elements that each story should contain.

For example, Marie Pavlovska defined storytelling as "an interactive form of communicating a story; it is an art form that is based on language, the use of voice, movement and gesture to portray elements, images and ideas based on a specific story to a specific audience". This definition emphasizes the use of the narrator's voice, movement, and gestures, which are the main elements of oral story delivery. According to Pavlovska, two basic types of storytelling can be distinguished: "oral" and "digital". Oral storytelling belongs to the improvisational arts. The narrator recounts the story, and the audience has to engage their own imagination. Digital storytelling is a form of storytelling using media. Storytellers create a digital story using multimedia tools including graphics, video, animation, music, and sounds. Both types of storytelling can serve to entertain, educate, preserve culture, instill knowledge, and value practices [11]. Carolyn Miller also describes how to create engaging and interactive stories for education, training, and promotion. She writes that the way the story is told has significantly changed in recent years. Stories are told through video games, interactive books, and social media [15].

Monika Nevolova [16] explains why storytelling is a powerful tool for influencing, learning and communicating with people. She states that not only words, but also the emotions the listener perceives when telling a story, help to understand and remember certain ideas and situations. The book introduces a simplified structure to help the storyteller create a simple story. Each story should have a chronological beginning (the concept), middle (the action), and end (the outcome, and what follows). In the context, it is important to familiarize the listener with the time and place in which the story takes place, who the main character is, what he or she wants to accomplish, and what he or she must overcome.

If the hero does not enter any conflict, the story becomes uninteresting. At the end of the story, the listener wants to know how it all turns out. Wright et al. place, among the basic elements of storytelling that create an interesting story, roughly the same elements as Nevolova does: setting, theme, characters, plot, and conflict [17]. Lisenbee and Ford introduce additional elements that help create compelling stories. These include a sense of humor, repetitive lines, a limited number of characters, age-appropriate language, and colorful pictures and illustrations [18]. Robert Pratten offers similar advice in his book. The key to success is to create an appropriate and interesting story. Each story must have a hero embedded in a setting that is familiar to the audience. The story should surprise and contain wit [19].

### 2.2. Digital Storytelling

Digital storytelling, abbreviated as DST, was first used in 1980 by theatre practitioners at the Center for Digital Storytelling in California, USA, to record, produce, and present stories [20]. The Center for Digital Storytelling has defined DST as a form of storytelling in which any individual can document and share a meaningful life experience, thoughts, or feelings by creating a short story using digital technology [20]. However, there is no single definition of DST, and since DST can take different forms, different explanations of the term can be found in the literature. In their article, Kajder et al. describe DST as *"a group of still images combined with a narrated soundtrack through which a story is told"* [21]. Davis and Weinshenker wrote in their book that a DST is *"a short, usually personal story, told in the first person and presented as a short film for viewing on television, on a computer monitor, or projected onto a screen"* [22]. From these definitions, digital storytelling can take the form of a video, an animated film, or a presentation of photographs accompanied by music and a narrator's voice. Technology has advanced to the point where stories can be converted into slideshows of photos, animations and videos using video cameras, cameras, and mobile phones [23]. Today, a wide range of tools and applications are available that allow DST at a very good level while requiring minimal knowledge in their use [24,25]. Today's teens are comfortable creating stories electronically and sharing them online using social networking tools, such as Facebook, YouTube, and MySpace.

The actual creation of a digital story, regardless of the type of DST, requires knowledge on multiple areas. The author must have at least basic IT user skills, a good knowledge of the story topic, and language competence. Stephanike Evergreen [26] points out four basic areas that need to be addressed. These are graphics, text, color, and layout. The recommendations are based on studies of research intelligence, cognitive psychology, communication, and graphic design.

### 2.3. Educational Digital Storytelling

Storytelling is an art that people have been practicing since time immemorial. Therefore, there are plenty of literary resources available. However, the use of digital storytelling in education, abbreviated as "EDS", after the English name "*Educational Digital Storytelling*", is a relatively new method of teaching [6,27]. EDS is one type of Digital Storytelling (DST) that is focused on education. The structure of EDS is the same as that of other story types. All stories should have a concept, a main character, action and end [16,17].

Since the 1990s, EDS has become a popular educational method in the USA and has gradually begun to be used in Asian and European countries. For example, as early as 2002, the authors of [28] proposed three ways to support learning with digital stories. Digital stories can be used as examples of concepts or principles that are taught through direct instruction. Second, they can be used as problem examples for students to solve. Third, stories can be used as hints for students to help them learn to solve problems. Helen Barrett found that digital storytelling facilitates the incorporation of four learning strategies from instruction: student engagement, reflection for deeper learning, project-based learning, and the effective integration of technology into instruction. In her 2006 article, she proposed a research project to collect data on the use of EDS in schools. Barrett reports that if the tool is

to be used routinely in practice, information on its impact on student learning, motivation, and engagement is needed [29]. Over the following years, many books and articles have been published on the topic of carrying out and using EDS, but only some of them describe the research methodology and concrete results of pedagogical experiments aimed at using the EDS method.

Research focused on the use of the EDS method in education has been carried out in various subjects in primary schools [30–32], secondary schools [10,33], and in universities [34,35]. Studies were, for example, conducted on the subjects of literature [36], foreign language teaching [12,31,37–39], ethics [13], or information technology [40]. Most sources report that the use of digital storytelling in education has positive results. Digital stories presented by teachers help students to better understand theoretical concepts and relate them to real life. They show interconnections within a larger body of learning and facilitate discussion of the topics presented in the story [41]. The authors of [31,42] state in their article that creating educational stories is very time-consuming but highly motivating. The multimedia elements of digital storytelling keep students' attention and increase their interest in learning [43]. For instance, some authors, in their article, report that the students who were surveyed on foreign language learning showed an increasing trend in their extrinsic and intrinsic motivation to learn and demonstrated a high interest in contributing their thoughts during digital storytelling activities [44,45].

On the other hand, some authors draw attention to the risks of using EDS. In 2016, Mateo Stocchetti developed a compendium in which fifteen experts addressed the use of digital storytelling in education. The authors pointed out that the role of EDS in teaching is often greatly overestimated. There are good points, but also downsides to digitalization in education. They are rather critical of the use of digital storytelling in education and warn educators of the risks of turning traditional narratives into digital stories. The use of information technology and media brings many opportunities, but also risks. Based on research and field experience, the authors provide ideas, evidence, references, and inspiration on how to use digital storytelling in the right way [46].

Brian Belland, who tested the use of EDS in STEM education, reports that the greatest impact was seen in students' cognitive outcomes. The author notes, however, that this statement may not be true for all subjects because the survey was only conducted on STEM subjects, i.e., science, technology, engineering, and mathematics [7].

Only one study on the use of EDS in teaching an economics course was conducted. In 2019, Lestari, Siswandari, and Indrawati created digital stories and investigated their impact on the flow of instruction and improved student achievement in economics in a secondary vocational school. The research results show that the development of educational media and the use of digital stories creates a more interesting and fun learning atmosphere, and that the students were more motivated to learn economics [10]. However, the study outcomes were not examined during this study.

One of the important sources for the design of the methodology of the pedagogical experiment was the research conducted by the authors Yang and Wu when they investigated the effect of EDS on the academic results, critical thinking and motivation of high school students to study foreign languages [31]. The study involved a total of 110 10th-grade elementary school students, which corresponds to the age of students in their first year at secondary schools in the Czech Republic. The EDS method was used in two English classes per week for one year. The independent variable was English instruction integrated with information technology at two different levels—lectures (control group) and EDS (experimental group). Both quantitative and qualitative data were collected during the study, including English language performance, critical thinking scores, responses to a questionnaire for learning motivation, as well as interview records of the students and teachers to assess the effectiveness of EDS in learning. Descriptive analysis, analysis of covariance (ANCOVA), multivariate analysis of covariance (MANCOVA), and qualitative content analysis were used to evaluate the collected data. The results of the study show that the EDS participants performed significantly better than the lecture participants did

amongst all three criteria studied—English proficiency, critical thinking, and motivation to learn. The interview results highlight the important educational value of EDS, as both the instructor and students reported that digital stories increased students' understanding of course content, willingness to explore, and ability to think critically about factors that are important in preparing students for the ever-changing 21st century [31]. This study is similar to the proposed pedagogical experiment in terms of the methodology used, and the age of the students. The curriculum and the intensity of the use of the EDS method are different.

Ge in his study compared the effectiveness of EDS and memorization of vocabulary learning in a group of adult Chinese e-learners [47]. A total of 60 students participated in the study, 30 in the experimental group and 30 in the control group. The students in the experimental group were taught using the EDS method and those in the control group were taught using memorization in one lesson which lasted 30 min. Two post-tests were administered to both groups, one immediately after the end of the class and the other three weeks later. The results of the data analysis showed that the EDS method was more effective than memorization, both in terms of short-term and long-term recall. In this study, the EDS method was used once, only in one lesson, similarly to the proposed experiment. The similarity can also be found in the evaluation of study results using the post-test, which takes place immediately after the lesson.

Previous research shows that the EDS method yields positive results on student motivation in humanities, social sciences, and STEM subjects. However, the effectiveness of this method on student achievement in the subject of economics has not been proven according to available sources. The aforementioned research in economics [10] focused on the implementation of EDS, its integration into teaching, its contribution to students' motivation, and the atmosphere during teaching. However, students' knowledge was not tested. The pedagogical experiment, which is described in the next section of this study, mainly focuses on the effect of the EDS method on learning outcomes in the subject of economics.

### 3. Data Methodology

The aim of this pedagogical research was to examine whether or not educational digital storytelling contributes to better study results in the subject of economics. First, a pilot phase of the research was conducted to validate the testing methodology. A total of 82 business academy students in the Czech Republic aged 15–19 participated in the pilot phase. The students were tested on economic topic "*Total and marginal utility*" using EDS and teacher's explanations supported by a presentation. The pilot phase of testing produced similar results to those of the pedagogical experiment. Several changes in methodology were made after the pilot phase. The total of six questions in the pre-test and post-test was increased to 16 questions due to the higher reliability of the test, and the total of six questions in the questionnaire survey was increased to 10 questions. Unlike the pedagogical experiment, the pilot phase was conducted in person. The pedagogical experiment was conducted online due to COVID-19, and at the same time it was possible to provide EDS testing and teacher-assisted presentation by the same teacher to all students in different schools.

The research hypothesis of the pedagogical experiment was defined similarly to that in the pilot phase (H1: *The use of educational digital storytelling leads to an increase in the individual students' results*). In the first phase of the research, study materials on the topic of price elasticity of demand and supply were prepared. A business topic was incorporated into the digital story called "*How Vojta sold shoes*". The short video, lasting seven minutes, showed the basic principles of the market mechanism and the elasticity of demand using an example of the young businessman´s mistakes. The EDS intervention and the PowerPoint presentation were created by one of the authors of the article, Jana Nunvarova, who teaches economics at the business academy. The EDS intervention and the presentation were verified by experts. Figures 1–3 show the screenshots of the digital storytelling.

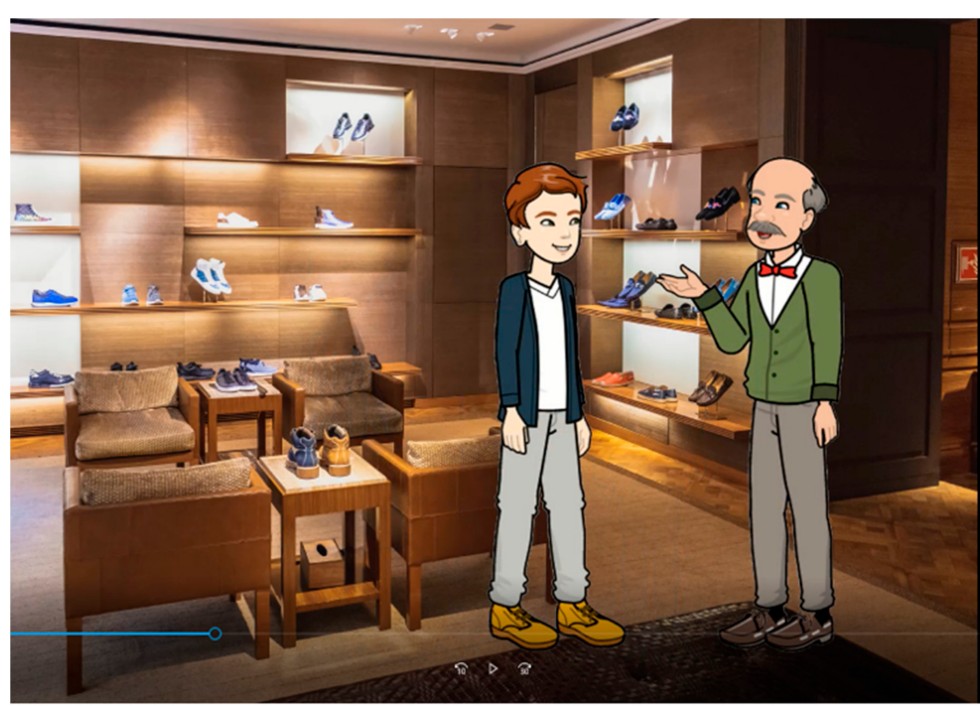

**Figure 1.** Screenshot of the digital storytelling called "*How Vojta sold shoes*".

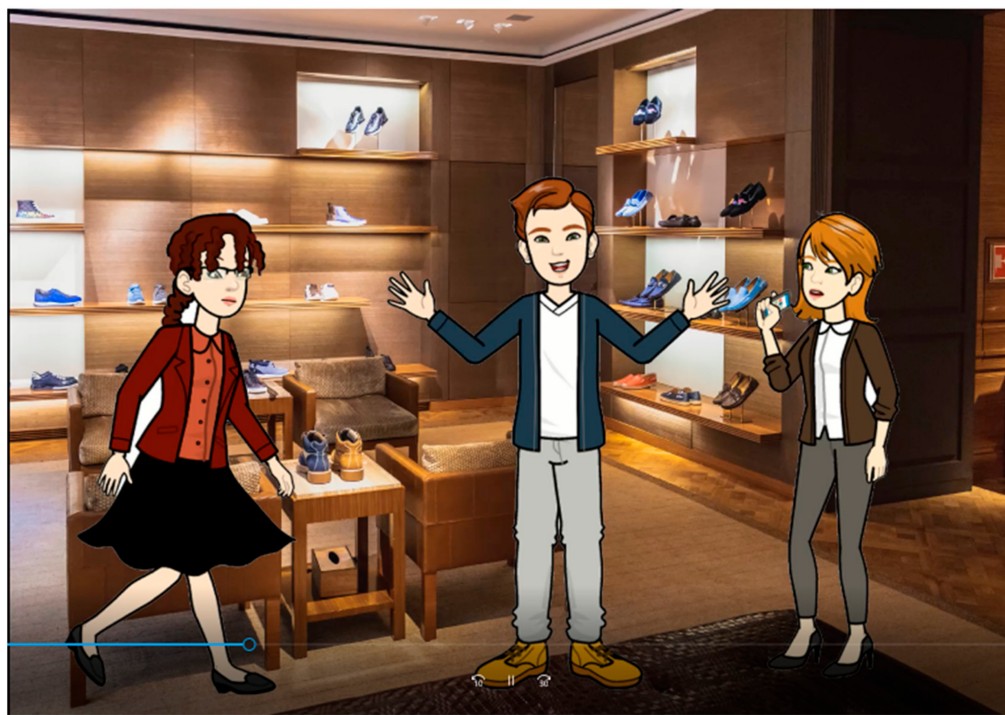

**Figure 2.** Screenshot of the digital storytelling called "*How Vojta sold shoes*".

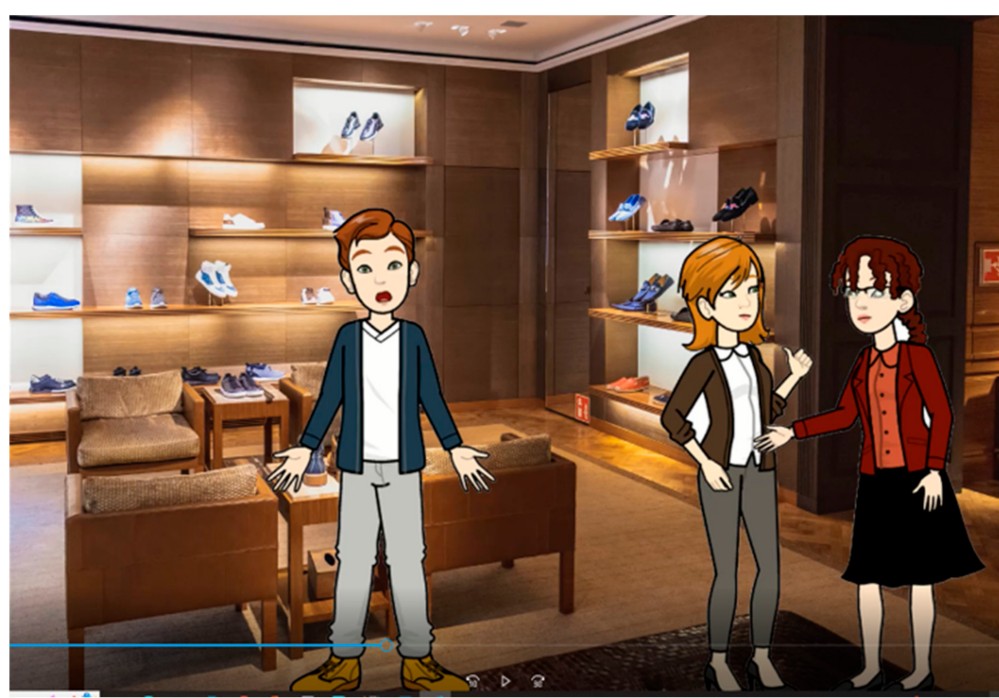

**Figure 3.** Screenshot of the digital storytelling called "*How Vojta sold shoes*".

A six-minute PowerPoint presentation was prepared on the same topic, "Supply, Demand and Price Elasticity", with an additional explanation given by the teacher. Figures 4 and 5 show screenshots of the presentation.

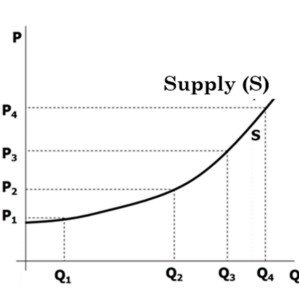

**Figure 4.** Screenshot of the presentation.

To verify the students' understanding of the given topic, a test with 16 closed questions was conducted. The same questions were used in both the pre-test and post-test. To assess the reliability of the test, the concept of the internal consistency of the test was used, indicating the consistency of the set of questions when creating summation indices. In this case, the meaningful relationships of the items that were to enter the index were verified. In the case of the index consisting only of dichotomous items, the most commonly used indicator of the internal consistency reliability was the score obtained from the Kuder–Richardson formula Formula 20, also known as KR-20. Mares, Rabusic, and Soukup stated

that a score higher than 0.7 should be considered consistent and reliable [48]. Having in mind the KR-20, the score of 0.714 was interpreted in the post-test, which indicated internal consistency and reliability. Furthermore, a questionnaire, focusing on the students' perception of the teaching methods and their motivation and priorities in the process of education, was prepared.

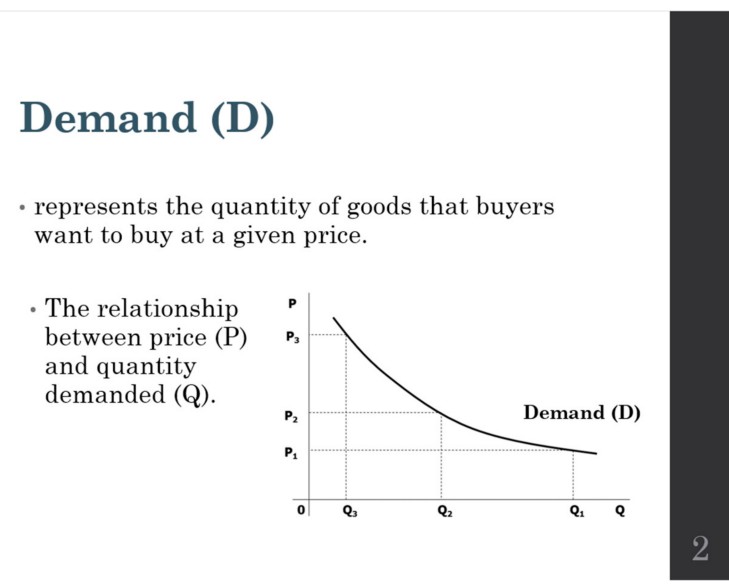

**Figure 5.** Screenshot of the presentation.

In the second phase of the research, the research participants were tested. Six business academies from the Hradec Kralove and Pardubice regions of the Czech Republic took part in the research. A total number of 856 secondary school students, aged between 15 and 19 years, were tested. Group N, consisting of 856 students, was randomly divided into two independent groups. In each participating school, students were randomly assigned by lot by the economics teacher. The experimental (n1 = 430 students) and control (n2 = 426 students) groups included students from all 6 schools and students from all grades. Testing took place online during distance learning via MS Teams, from February to June 2021. Thanks to the online teaching using MS Teams, the same conditions were ensured for all respondents during the testing. All respondents were familiarized with the testing process in the same way, watched the same digital storytelling and listened to the same teacher's explanation supported by a presentation. Students in the experimental and control groups were tested separately for the duration of a class, i.e., 45 min. First, students from both groups were introduced to the testing process and completed a pre-test. The time limit for the pre-test was 10 min. Then, 460 students from the experimental group (n1) watched a digital storytellingcalled "*How Vojta sold shoes*" focused on the economic topic of supply, demand and price elasticity. Control group n2, with 426 students, used the traditional teaching method. The topic "Supply, Demand and Price Elasticity" was explained to them by the teacher with the support of the presentation. The time limit for viewing EDS and the teacher's explanation was 7 min. Both groups then completed the post-test, consisting of 16 closed questions. The questions in the post-test were identical to the questions in the pre-test. The time limit was again 10 min. Another 7 min of testing was intended for the exchange of teaching methods in both groups. The students of the experimental group were given explanation of the topic of supply, demand and price elasticity by the teacher with the support of a presentation, while the students of the control group watched an EDS example. As a result, students in both groups were able to compare and evaluate both teaching methods in the final questionnaire survey. The methods and their effectiveness in teaching were evaluated on a Likert scale. Students could also write

their opinions on the course of testing and priorities in teaching methods. In order to analyze the obtained data, IBM SPSS Statistics software 26 was used.

## 4. Results

As has been already mentioned above, the students' results were measured by the pre-test and post-test and their values were recorded. The results were calculated using IBM SPSS Statistics software. A significance level of 5% was considered when performing the tests. The students' initial knowledge of the topic dealing with the price elasticity of demand and supply was examined through the pre-test consisting of 16 closed questions. The results according to both groups are shown in Table 1, in which the students' success is reported in the points obtained from the pre-test, and Figure 6 below.

**Table 1.** Descriptive statistics of the experimental and control groups in the pre-test.

|  | Group | Mean | 95% Confidence Interval for the Average | | Median | Std. Deviation | Minimum | Maximum |
|---|---|---|---|---|---|---|---|---|
|  |  |  | Lower Limit | Upper Limit |  |  |  |  |
| Pre-test | Experimental | 9.86 | 9.64 | 10.08 | 10.00 | 2.303 | 3 | 16 |
|  | Control | 10.27 | 10.04 | 10.50 | 10.00 | 2.393 | 2 | 16 |

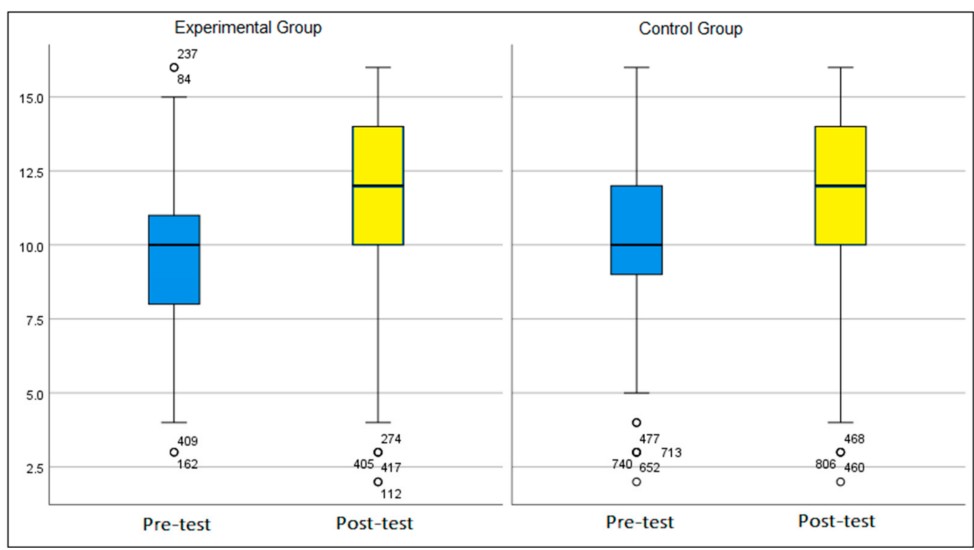

**Figure 6.** Box diagram of the pre-test and post-test results for the experimental and control groups.

The research hypothesis was defined as follows:

**H1.** *The use of educational digital storytelling leads to an increase in the individual students' results.*

After the EDS intervention and the intervention using the traditional method of teaching (teacher's explanation with the support of the presentation), the post-test was used in order to detect the differences in the students' results. The descriptive statistics and the box diagram of the post-test results for both the experimental and control groups are show in Table 2 and in Figure 6.

An analysis of variance with repeated measures (ANOVA) could be performed using the time/test factor, whose values were pre-test and post-test, and the group factor, whose values were those of the experimental and control groups, as two independent factors [49,50]. Thanks to the method of measurement, where the measurement was repeated on the selected research sample/students, the time/test factor could be considered a factor with a within-subject effect, and a group factor that divided the population could be a factor expressing an intergroup effect (between-subject effect). The ANOVA for repeated



measures confirmed the effect of time/test (F (1.854) = 266; *p*-value < 0.001 and partial eta squared, 0.238); thus, there was an increase in the average score in the post-test. The group influence was not statistically significant (F (1.854) = 0.887; *p*-value, 0.347; partial eta squared, 0.001), i.e., it did not matter to which group, experimental or control, the student belonged. The calculation also showed the interaction between the group factor and the time/test factor (F (1.854) = 8.287; *p*-value = 0.004 and partial eta squared = 0.010). The values given in Tables 1 and 2 mean that students from the experimental group had a higher average success rate in the post-test than the students from the control group did. Despite a consideration of the differences between the results in Tables 1 and 2, and the partial eta squared of 0.010, this improvement is very small. However, this result represents support for the hypothesis 1/H1. The overall development of the average test results is illustrated in Figure 7.

**Table 2.** Descriptive statistics of the experimental and control group in the post-test.

|  | Group | Mean | 95% Confidence Interval for the Average | | Median | Std. Deviation | Minimum | Maximum |
|---|---|---|---|---|---|---|---|---|
|  |  |  | Lower Limit | Upper Limit |  |  |  |  |
| Post-test | Experimental | 11.61 | 11.33 | 11.89 | 12.00 | 2.906 | 2 | 16 |
|  | Control | 11.50 | 11.23 | 11.77 | 12.00 | 2.843 | 2 | 16 |

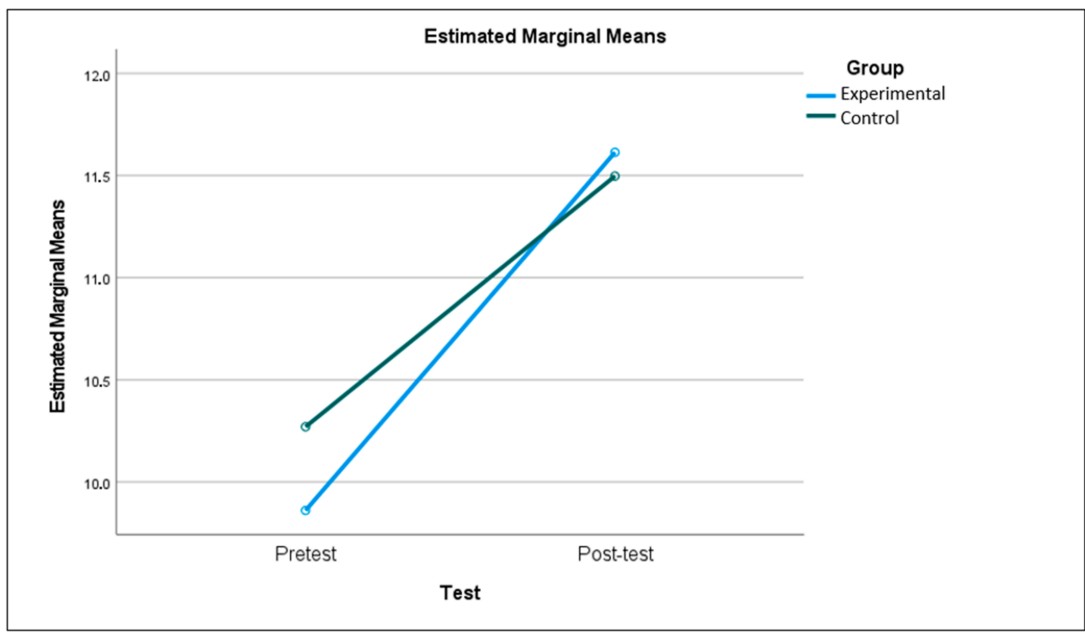

**Figure 7.** The overall development of the average test results for the experimental and control groups.

Because the repeated measures ANOVA showed a significant interaction, we now introduce the results of a post hoc comparison of groups. First, we dealt with pre-test. Based on the one-sided two-sample *t*-test with equal variances (Levene's test; F = 0.697, *p*-value = 0.404), we can conclude that the mean of the experimental group is significantly lower than the mean of the control group (t = −2.551 with d.f. = 854 and *p*-value = 0.005). This means that the students of the experimental group had slightly worse average results from the pre-test, cf. Table 1. Secondly, we deal with post-test. Based on the two-sample *t*-test with equal variances (Levene's test; F = 1.249, *p*-value = 0.264) we cannot conclude that there is a significant difference between the mean of the experimental group and the mean of the control group (t = 0.586 with d.f. = 854 and *p*-value = 0.558). This means that the students of the experimental group and the students of the control group had almost the same average results, cf. Table 2. Given that the students of the experimental group started with

worse results, this indicates that they made more progress than students of control group did. Finally, we introduce the results of the paired *t*-tests to compare the individual progress of students. Starting with the students of the control group, the mean of the individual differences between the post-test and pre-test is 1.228, and based on the pair *t*-test, we can conclude that this difference is significant (t = 9.501 with d.f. = 425 and *p*-value < 0.001). The similar results can be observed for the students of the experimental group. The mean of the individual differences between the post-test and pre-test is 1.753, and based on the paired *t*-test, we can conclude that this difference is significant (t = 13.584 with d.f. = 429 and *p*-value < 0.001).

At the end of the research, after using both EDS and the standard method of teaching, the questionnaire survey was conducted in order to understand the students' perception of the teaching methods. Most students rated EDS very positively and found it beneficial in the process of learning economics. They considered EDS more interesting, in comparison with the standard teaching method, and claimed it was easier to remember the new technical terms when using it. Despite this fact, 10% of the students from the experimental group did not prefer the EDS method to the standard teaching at all, and 44% of the students would rather have not used it. On the contrary, the control group, which was first introduced to the new topic through standard teaching, and only watched an EDS intervention after the post-test, would have rather used the EDS method in the process of education. A total of 45% of the students answered that they would prefer EDS to the standard teaching method, and 11% of the students from the control group claimed to definitely prefer it. A total of 61% of the students from the experimental group and 63% of the students from the control group said they would prefer the combination of both methods in learning economics.

## 5. Discussion

The aim of the pedagogical experiment was to discover whether or not EDS contributes to better study results in the business subject taught at high school. The research was conducted from February to June 2021 as part of the online distance teaching of economics at six business academies in the Czech Republic. A total number of 856 students (15 to 19 years old) were tested. The students were randomly divided into two independent groups. There were 430 respondents in the experimental group, using the method of digital storytelling in teaching economics. The control group, consisting of 426 students, was introduced to the same business topic through the teacher's explanation with the support of a PowerPoint presentation.

By comparing the results from the pre-tests and post-tests obtained from the experimental and control groups, it can be statistically proven that at the 5% significance level, there are significant differences between the results of the experimental and control groups. This means that hypothesis 1/H1, "*The use of educational digital storytelling leads to an increase in the individual students*' results", was not rejected. The students from the experimental group reached higher average success in the post-test than the students from the control group did. Even though the improvement was very small, it supports hypothesis 1. If the learning outcomes with another subject are compared, for example, with English language learning, the benefits of EDS are far more evident in this subject. The results of the study [7,31,39,47] show that EDS participants performed significantly better than lecture participants did. In the research of [31], the reason may be the fact that the English language instructions using digital storytelling were given regularly twice a week for one year. Other factors of the effectiveness of EDS are the difficulty of the given topic, the scenario, and the treatment of digital storytelling, but also the quality of the teacher's explanation, etc.

According to the available sources, the only research focused on economics was conducted by Lestari, Siswandari, and Indrawati [10]. This research was focused on the implementation of EDS and students' evaluation of this method. Regarding this, the student outcomes obtained from this study cannot be compared with the results of the pedagogical experiment.

The results of the questionnaire survey, focused on the students' motivation and their opinion of the methods used during the research period, confirmed the results obtained from the previous research [6–9,43]. The majority of students rated the digital storytelling method very positively and considered the use of EDS beneficial. They found teaching more interesting and fun. For 66% of the students, the concepts explained through digital storytelling were more understandable and easier to memorize. In addition, most students preferred to combine the EDS method with a standard approach to teaching. Furthermore, the research reveals that an explanation of the new topic provided by the teacher is considered necessary in the process of economic education. Regarding this, the research participants perceive the EDS method as an interesting additional tool that provides practical examples in order to help students to understand the subject matter better. The obtained results agree with the survey results by Robin [41]. EDS shows interconnections within the broader curriculum and facilitates discussion of the themes presented in the story. However, teacher interpretation is irreplaceable in some subjects and topics.

An advantage of this pedagogical experiment is the large number of respondents. All the participants of the research attended the business academy at the time of the research period. They all knew the basic economic concepts, but it is important to mention that the chosen topic had not been discussed before the pedagogical experiment was conducted. It could therefore be assumed that their initial knowledge of the issue was at the same level. The whole test took place during the period of online distance learning. The same testing conditions were maintained. The students completed the pre-tests and post-tests using the MS Forms application. Through MS Teams, they were given the same digital storytelling intervention and explanations by the same teacher with the support of a presentation.

There are some limitations of this study. Firstly, the conclusions of the pedagogical experiment refer to only a selected sample, and although the sample consisted of 856 students, the results cannot be generalized. The pedagogical experiment was conducted only once, and further investigations with different topics in economics are necessary to verify the results. The digital storytelling intervention and the presentation were short so that the entire test could be carried out within the duration of one class. The question is whether or not a longer EDS intervention would have helped students understand the economic topic better than a standard lesson would. Based on a one-sided two-sample *t*-test with an equality of variances, it was found that the mean of the pre-test scores of the experimental group's students was significantly lower than the mean of the control group. Although the assignment of the respondents into groups was random and the topic was not discussed in school, the experimental group students had slightly worse mean scores on the pre-test (Table 1). Focusing on the post-test, the results of the experimental and control groups were almost the same (Table 2). Since the students of the experimental group started with worse scores, it can be said that they made more progress than the students of the control group did, which supports hypothesis 1/H1. However, the difference in the pre-test scores of the experimental and control groups is another reason why the results obtained cannot be generalized and a re-investigation would be advisable. Other factors that could have influenced the study results are the choice of the story and its treatment, the relatively high success rate of the students in the pre-test for both groups, and the choice of the concepts explained. The results of the experiment could have been influenced by the author who created the EDS intervention and the presentation, the teacher's interpretation, and the atmosphere in the classroom. Another question is how the results of the pedagogical experiment were influenced by online instruction. The importance of face-to-face teaching and the possibility of direct communication with the teacher was confirmed by some authors [7,35]. Online teaching was an advantage for maintaining the same testing conditions, but at the same time was a disadvantage. There was no discussion after watching the video or after the teacher's explanation. Further research could focus on determining the effectiveness of EDS if the testing process was supplemented with discussion among students and teaching was conducted face-to-face.

The data obtained could be subjected to a more thorough examination using other statistical methods, such as regression and correlation analysis, which would enable a search for relationships between the results and students' attitudes towards the EDS method with respect to students' gender, field of study, or age. Interesting results could also be found when comparing the success rates of different questions of the test because some questions were based on definitions, and others were focused on practical examples and the understanding of theory.

The aim of the pedagogical experiment was to investigate whether or not digital storytelling contributes to better learning outcomes in business courses taught in high school. At the end of the test, a questionnaire survey was conducted to find out the students' views on the EDS method in teaching economics. Most students preferred a combination of both methods. Further investigation could be focused on how effective the combination of both methods is during the teaching process. The differences in learning outcomes after using one method and after combining both methods could show the importance of incorporating different teaching methods.

## 6. Conclusions

The pedagogical experiment and previous research show that the use of educational digital storytelling has a positive effect on students' motivation and learning atmosphere. Students appreciate the connection between theory and practical examples. Real-life stories linked to the topic help students understand theory better and engage in discussion. The question is whether or not the EDS method produces better learning outcomes than does the standard method of a teacher's explanation with the support of a presentation. As stated by the authors of [7], the EDS method may not be suitable for all disciplines. Some authors [46] point out that the role of EDS in teaching is often highly overestimated.

In the subject of economics, which was the focus of the pedagogical experiment, the students rated the use of the EDS method in teaching very positively, and the learning out-comes from the students of the experimental group reached higher average success in the post-test than those from the students of the control group. Hypothesis 1/H1, "The use of educational digital storytelling leads to an increase in the individual students", was not rejected based on the obtained results. Nevertheless, according to the students who participated in the pedagogical experiment, teacher explanation is essential for them in the subject of economics. They perceived the digital story as a motivational tool and a method to support their learning.

**Author Contributions:** Conceptualization, J.N. and P.P. (Petra Poulova); methodology, J.N., P.P. (Petra Poulova) and P.P. (Pavel Prazak); software, J.N. and P.P. (Pavel Prazak); validation, J.N., P.P. (Petra Poulova) and P.P. (Pavel Prazak); formal analysis, J.N.; investigation, J.N.; resources, J.N.; data curation, J.N. and P.P. (Pavel Prazak); writing—original draft preparation, J.N. and B.K.; writing—review and editing, J.N. and B.K.; visualization, J.N. and P.P. (Pavel Prazak).; supervision, P.P. (Petra Poulova); project administration, P.P. (Petra Poulova); funding acquisition, P.P. (Petra Poulova). All authors have read and agreed to the published version of the manuscript.

**Funding:** This work was supported by the SPEV project 2105, Faculty of Informatics and Management, University of Hradec Kralove.

**Institutional Review Board Statement:** The Committee for Research Ethics at the University of Hradec Kralove has approved this research proposal. The described study procedure is in accordance with the Ethical research framework of the Ministry of Education, Youth and Sports in the Czech Republic and ethical requirements in research.

**Informed Consent Statement:** Informed consent was obtained from all subjects involved in the study.

**Data Availability Statement:** Data can be made available on request from the first author.

**Conflicts of Interest:** The authors declare no conflict of interest. The funders had no role in the design of the study; in the collection, analyses, or interpretation of data; in the writing of the manuscript; or in the decision to publish the results.

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
