# Peer review of "Effectiveness of Digital Storytelling in Teaching Economics"

_education, doi:10.3390/educsci13050504_

Round 1
Reviewer 1 Report
The study examines whether the storytelling method is more efficient and effective in digital education than the traditional teacher's explanation. The question raised is topical and important, but too narrow.
Although the study examines the effectiveness of the two types of methods in terms of student motivation, interest and satisfaction, it did not develop a hypothesis for these questions and did not use statistical methods to validate the results.
In order for the research results of the study to be valuable enough for the development of the teaching methodology, a scientific examination of the above facts is necessary.
In addition, it would have been worthwhile to examine how effective the combination of the two methodologies is during the teaching process. I assume that the story telling and the teacher's explanation together are the most effective.
Author Response
Cover letter of the revisions
- It was added in the abstract on line 15 that the data were collected at six business academies "in the Czech Republic".
- Correction of typo in the word “behaviour” on line 34.
- Lines 49-53 have been removed to reduce repetition of information.
- Lines 53-59 have been retained to provide additional sources, which are referenced below.
- Lines 60-64 have been retained to introduce Hypothesis 1, the results of which are briefly described in the following paragraph.
- Lines 64-77 have been removed to reduce repetition of information.
- Correction of a typo - addition of a comma on line 234.
- Section 3 has been reworded as "Data Methodology".
- Information on the pilot phase of the pedagogical experiment, including the number of respondents, results, and changes in methodology, has been added to Section 3, lines 262-276.
- The research hypothesis of the pedagogical experiment was written to Section 3, lines 277-279.
- The EDS author and presentation, including information on expert verification has been added on lines 284-286.
- Added three screen shots of the digital stories to Section 3.
- Added two screen shots of the presentation to Section 3.
- Figure descriptions 1-5 have been added.
- Information has been added to line 302 regarding the identity of questions in the pretest and posttest.
- Correction of typo in the word “Kralove” on line 315.
- Changing the numbering of Figure from No. 1 to No. 6
- Table number 2 has been uploaded again in its original form. It was accidentally moved and deleted when inserting multiple images.
- Changing the numbering of Figure from No. 2 to No. 7
- Changing the numbering of Figure from No. 3 to No. 8
- Correction of a typo - addition of a comma on line 467.
- A description of other limitations of the pedagogical experiment has been added to the Discussion section on lines 500-504. Testing was conducted only once using a short EDS versus a PowerPoint presentation.
- Lines 504-506 have been removed to reduce repetition of information.
- A description of other limitations of the pedagogical experiment has been added to the Discussion section on lines 509-510. The results of the experiment may have been influenced by the EDS author and presentation.
- A topic for further investigation focusing on the combination of both methods in teaching was added to the Discussion section on lines 526-533.
- Correction of typo in the word “Committee” on line 563.
Correction of typo in the word “Kralove” on line 56
Reviewer 2 Report
This is an interesting and timely manuscript but there is room for improvements as shown below:
In the abstract section, it would be useful to state the country of the data collection.
Some of the information in the Introduction was presented again in later sections. Suggest to remove line 49-77 to reduce repetitions.
Section 3 would be better to reword it as the “Data methodology” section. It would be useful to include some screen shots of the digital stories as well as the Powerpoint. It was good to have performed a pilot test first but what other information was collected and what changes have been made. How many questions were asked in the pre- and post-tests?
Although many statistics were performed, this study has more limitations and future research than stated, e.g., just one experiment using a short digital story as compared with Powerpoint presentation was too brief to draw any meaningful conclusion. BTW, who has created the digital story and the PowerPoint which could have an impact on the experiments. The authors are suggested to think deeper and wider when revising the paper. Furthermore, some typos are also found.
Author Response

(The authors gave the same response as above.)

Round 2
Reviewer 2 Report
Congratulations! The revised paper looks good to me.
Author Response
Thank you very much